# Impact of Screen Time on Children's Development: Cognitive, Language, Physical, and Social and Emotional Domains

**Vaishnavi N. Panjeti-Madan * and Prakash Ranganathan**

School of Electrical Engineering and Computer Science (SEECS), University of North Dakota,
Grand Forks, ND 58202, USA
* Correspondence: vinni.vaishu@gmail.com

**Abstract:** Technology has become integral to children's lives, impacting many aspects, from academic to socialization. Children of today's generation are growing up with digital devices, such as mobile phones, iPads, computers, video games, and smart gadgets; therefore, screen time has become ubiquitous in children's daily routines. This paper provides a review of screen time usage and its impact in children across multiple developmental domains: cognitive, language, physical, and socio-emotional domain of children under eight years of age. The cognitive domain considers factors such as attention span and memory; language domain examines vocabulary, speech, and language development; physical domain focuses on motor development, exercise, sleep, and diet; and social-emotional domain considers relationships, self-identity, and emotional behaviors/regulation. Our findings are mixed, as there are both benefits and drawbacks in technology use, but screen time in children requires controlled observation and monitoring for sustainable improved progress across developmental domains. Specific recommendations advise that children's screen time per day should be limited to zero minutes (min) (0–2 years), <60 min (3–5-years), and 60 min (6–8 years).

**Keywords:** media; screen times; child developmental domains

## 1. Introduction

Technology and media have become a vital part of the lives of infants, toddlers, preschoolers, and school-aged children [1,2]. In today's generation, digital devices, such as TVs, smartphones, tablets, digital toys, and computers, are present in every home [3]. Young children are being introduced to mobile gadgets and other smart devices during their early infancy via video communication by parents to communicate with distant family and friends [4,5]. Babies are subjected to background TV, while parents and siblings watch their shows on television [4]. Sometimes parents and caregivers use media and other devices as a calming tool for children [6]. Children of today's generation are growing up with technology such as mobile and smart devices in residential and academic settings [7]. In the US, about 75% of parents (with kids under eight years) possess a smartphone or any other touchscreen device [8]. The latest advancements in artificial intelligence can also impact young children in many ways. Language models such as ChatGPT can help children learn and write flawless essays; realistic image generation models such as DALL-E can also give thoughts to young creative minds [9]. The American Academy of Pediatrics (AAP), a professional association of pediatricians in the US, recommends that infants under two years should not be exposed to media [10,11]. However, babies are often introduced to a media device before 12 months of age. Often a two-year-old toddler can easily control the touchscreen technology device [12].

In this paper, we start by introducing the topic of screen time for children and its impacts. In the background section, we explore the historical context of screen time for children, how it has changed, the different types of screen time, and their impact on different child development domains. In the next section, we discuss our approach to

the systematic review of the existing literature on the impact of screen time on children's health and development. In Section 4, we dig deeper into the existing literature on the positive and negative impacts of screen time on children, including the influence of screen time on developmental domains. In Section 5, we discuss the impact of COVID-19 on children's screen time, including the increased use of screens for remote learning and socializing. In the following section, we highlight some practical recommendations for parents on minimizing their children's screen time from the existing research, highlighting the importance of setting boundaries, encouraging alternative activities, and modeling healthy screen use. Section 7 summarizes the recommendations for parents, teachers, and children from organizations such as the World Health Organization (WHO), AAP, and the American Psychological Association (APA). We conclude by summarizing the key findings from the literature review on the importance of limiting screen time for children and the role of parents in managing their children's screen use.

## 2. Background

"Technology" refers to the tools and equipment (devices) used for entertainment or informational purposes, and "media" refers to the digital content and games accessed on the devices [13]. Media before the 2000s was confined just to televisions for the most part. With the advent of modern technology, including smartphones, tablets, digital toys, and gaming tools, children are often occupied with digital content that comes in many sizes and forms. Excessive technology usage has many health, developmental, and behavioral challenges in children [14]. The different types of media existing in the present day are shown in Figure 1. Kids can access television, smartphones & tablets at home early on. Children of all ages can learn new skills using educational resources, such as educational videos and interactive programs, at home and school. Children use social media platforms, such as Facebook, TikTok, and Instagram to communicate and connect with friends. The usage of social media is widespread among tweens and teens. Video games are popular among all age groups of children, and they can access video games and other gaming tools at home [15].

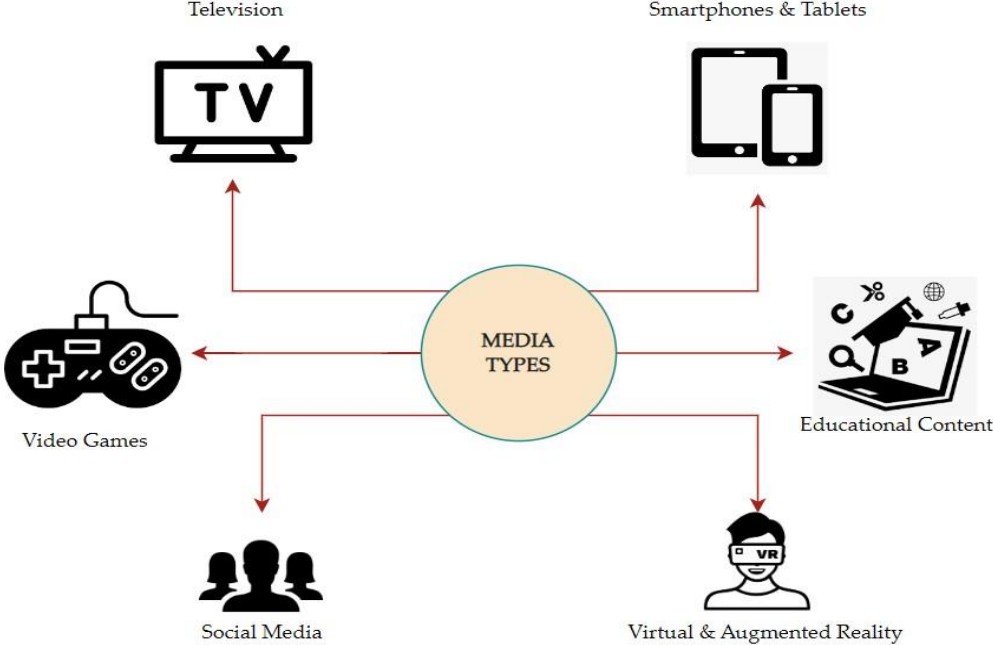

**Figure 1.** Different types of digital media.

The aggression depicted in video games can lead to violent behavior in young players, a significant concern that needs to be addressed, as players can imitate actions from the virtual world in real life [16]. Virtual reality (VR) can be an exciting and engaging experience

for children. However, excessive usage can lead to cybersickness, a motion sickness caused by exposure to virtual environments. Symptoms can include nausea, dizziness, and disorientation, negatively impacting a child's physical and mental health [17]. Technology and media usage among infants and toddlers is growing significantly [18]. A child's average daily screen time is drastically increasing with age. Children between 0–8 years use about an average of two and a half hours (h) of screen media daily. Television and watching videos online account for most screen time use [7]. A 2019 study in the U.S. reports that in the past two decades, children's media usage time increased by 32% [19]. Another research report showed that total screen time in children aged 0–2 years increased from 1.32 h to 3.05 h a day and 3–5 years from 2.47 h to 2.56 h a day between 1997 and 2014 [20]. According to research in the U.S., about 74% of parents report that their children under age two watch TV [21]. A national survey by Wartella revealed that in media-centric parenting, the average screen time for kids is 4.30 h a day. The report describes the relationship between media-centric parenting and screen time in children as directly proportional. More details on the parenting styles and the relationship with screen time are shown in Figure 2. This survey also shows that 50% of homes have the TV turned on most of the time, and 48% have a TV in the child's bedroom [2].

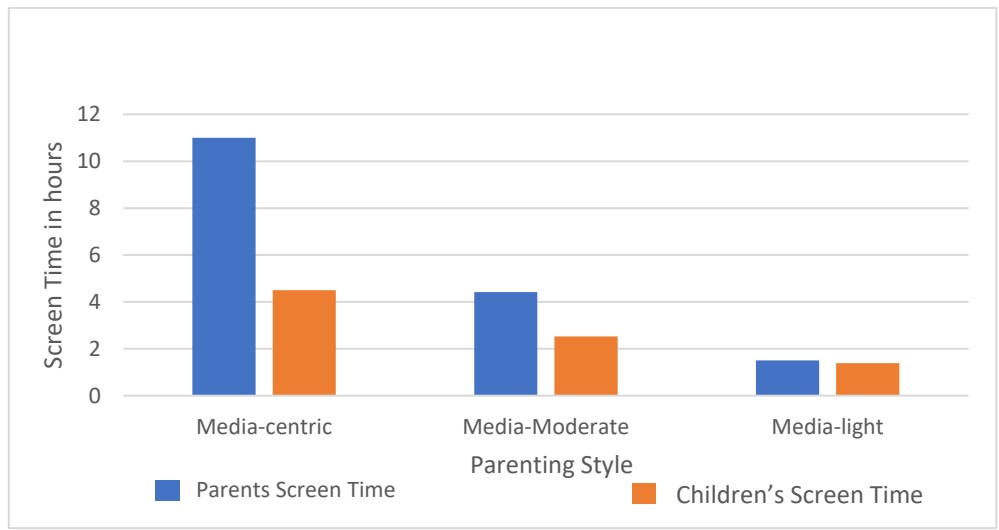

**Figure 2.** Parenting styles affect child screen time [2].

Media-centric parenting is a style in which a substantial portion of the child's daily routine involves using electronic devices, such as smartphones, tablets, computers, and televisions. Media-moderate parenting aims to balance a child's use of technology and other activities by limiting screen time and monitoring the content children consume. Alternatively, the media-light parenting style limits a child's exposure to electronic media. It sets boundaries and rules around media use, encouraging outdoor play, reading, and creative play [2].

Table 1 shows children's average media use time in different age groups.

**Table 1.** Average media usage time per day by different age groups of children [7].

| Age | Average Media Use (Hours/Day) |
| --- | --- |
| 0–2 years | 0.81 h (49 min) |
| 2–4 years | 2.30 h |
| 5–8 years | 3.05 h |

Table 2 shows the increase in children's recreational screen time, educational screen time, and total screen time during COVID-19 pandemic periods in comparison with prepandemic screen time.

**Table 2.** Change in children's screen time per day during the pandemic periods when compared to pre-pandemic screen-time rates [22].

| Screen Time (Type) | Screen Time per Day (in Hours) | | |
|---|---|---|---|
| | Pre-Pandemic | First Pandemic Period | Second Pandemic Period |
| Recreational Screen Time | 4 h | 4.89 h (+0.89) | 5.59 h (+0.70) |
| Educational Screen Time | 0.5 h | 1.43 h (+0.93) | 1.89 h (+0.46) |
| Total Screen Time | 4.4 h | 6.15 h (+1.75) | 7.26 h (+1.11) |

Media usage in younger children impacts their health and development [10,23], reduces babies' focus and attention span [24], sleep [25], physical activity [26], language development and communication [27,28], and socio-emotional health, and causes behavioral challenges [29,30]. Thus, a correlation exists between a child's screen time usage and development [31].

Many factors impact technology use in children. Household income, parental education, and family ecology (single parent) have been known to impact the media usage levels of a child [2,32]. In addition, ethnicity and race play significant roles in perceiving the educational benefits of media screen time. Some parents view learning, creativity, and social skills as vital motivating factors for child screen use. In contrast, others perceive that media adversely impacts children's physical activity. In the height of the COVID-19 pandemic, with online learning options, less social interaction, and limited outdoor activity, most children's lives revolve around media and technology [7].

Digital Technology is influencing children's lives. Media usage has several implications for a child's cognitive, language, physical, and socio-emotional development [14]. Different skills are clustered under different development domains [33]. A more precise description of the domains is provided in Figure 3.

**Figure 3.** The main childhood developmental domains [33].

### 3. Materials and Methods

Developed in 2009 by an international group of researchers, PRISMA (Preferred Reporting Items for Systematic Reviews and Meta-Analyses) is an approach that comprises a four-phase flow diagram that guides researchers through a systematic review. It helps from the early stages of study selection to data synthesis and reporting the results [34].

Our research question is, "How do technology and media impact children's development?" Using the PRISMA approach, we conducted a comprehensive search of relevant databases, using keywords and search terms such as "technology", "media", "children", "screen time", "media use", "child development", and "impact" across a range of different disciplines, including psychology, education, sociology, and child behavior. After reviewing the abstracts and titles of each article to determine whether it meets the criteria for inclusion in the review, some articles were excluded. Then, another screening level was performed by going through a full-text review of the articles to determine their relevance to the research question. After this systematic screening, relevant data concerning screen time from each article were extracted to tabulate findings across the studies. We also identified some key issues and guidelines to minimize the impact of technology and media on childhood development.

Building upon the available template for the PRISMA flow diagram, we present the different phases of our systematic review in Figure 4. The flow diagram visually represents the review process, highlighting the number of studies screened, assessed for eligibility, and included in the review, along with the excluded studies with the reasons for exclusion at each stage [35].

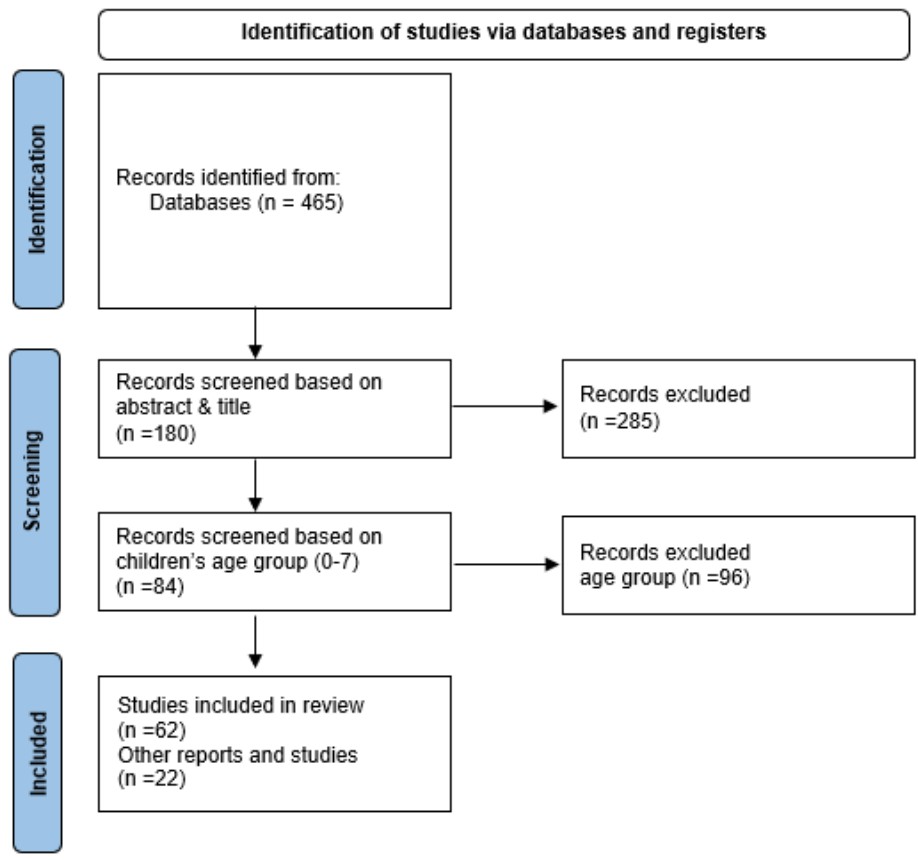

**Figure 4.** Different phases of our systematic review.

Technology use in younger children has been increasing over the past few decades. Our literature review focused on research articles since 2000. In this paper, we have included articles from Google Scholar, Science Direct, and Elsevier journals. A broad range of articles focused on the impacts of technology on young children. The search yielded

articles on different child development domains and empirical studies identifying media usage times and developmental impacts. Our focus was on summarizing positive and negative effects, recommendations, and best practices. The search results were refined by adding relevant search terms, and the results were limited to articles dealing with children below eight years of age.

## 4. State of the Review

Research studies describing the benefits and effects of technology on infants, toddlers, and young children are discussed in this section.

Rustem Mustafaoglu et al. [14] reviewed the literature on digital technology usage in children and how it impacts the child development. The review aims explicitly at the increasing usage of modern technologies such as smart televisions, tablets, computers, video games, digital toys, and smartphones. It also discusses various developmental, health, and other behavioral risks associated with long-term technology usage. The researchers of this study suggest that children under two years of age should not be exposed to any technology screens. In addition, the child's screen time per day should be restricted to 1–2 h only during the day. Parents need to be conscious of the harmful effects of technology and its usage; and need to set proper guidelines for the benefit of the kids.

Laura E. Levine et al. [32] have studied the use of media and technology (iPad or tablet, mobile phone, TV, MP3, iPod, laptop) by infants and toddlers (under 36 months). The study participants included 326 children (including infants and toddlers) and parents (primarily mothers), all U.S. residents from fall 2014 to spring 2015. Out of 326 infants, 190 were boys. Online surveys and questionnaires were part of the research. Parents were classified into an educational motive group and a non-educational motive group. The survey consisted of questions to parents to report and measure the self-regulation challenges in toddlers (how frequently the child is fussy, bored, wants toys, wants help to fall asleep) and explanatory questions to parents for reasons for providing technology devices to kids. The parents were asked to describe the frequency of technology used by children; how often kids use media together with parents, siblings, others, or alone; what types of content the kids watched (educational, non-educational, games, videos, movies, TV shows, Skype, or other communication); and at what age the child had first access to mobile or other technology. The parental media usage type, frequency, and time were also collected. The analysis of this study revealed that nearly 47% of infants were introduced to mobile technology before 18 months of age (between 7–18 months), and 40.7% of babies in this study sample were introduced to more than one technology device under 12 months of age. About 60% of the children used mobiles and tablets, and less than one-third used laptops, iPods, video games, and other media devices. Half (51%) of the children used technology to watch videos and movies, and 30% for video communication (Skype). In comparison, less than 25% used technology for educational purposes (reading books). The parents with high educational motivation participated with their kids in media usage. In contrast non-educational motivation and low parental education predicted increased levels of infant media and technology use alone. The study concluded that the family media ecology defines the time of technology used by children. Moreover, parents with non-educational motivation used technology devices to manage their kids' complex behavior and emotional regulation.

Sowmya Anand and Krosnick [36] assessed demographic elements that anticipate media usage amongst younger infants, toddlers, and children between six months and six years of age. The researchers conducted a nationwide (U.S.) telephonic interview of 1065 parents from April to June 2003. The questions included information about the child's media usage time on different kinds of technology such as TV (for watching videos), DVD, Xbox, hand-held video gaming tools, computers (for playing games or other purposes), mobile devices, and reading on the day prior to the interview. Demographic information collected in the study includes parental income, age, education, employment status, race, marital status of parent, as well as the child's age, gender, and native language, and the

number of households in a family are considered predictors to study kids' media behavior. The results of the research demonstrated that the child's age and race, and the parental education, employment, and marital status had a significant effect on the kids' technology usage and duration. In contrast, the gender of a child, primary language spoken at home, and parent's occupation had less influence on kids' media behavior. Significantly the parents' income did not affect media use in children.

Sundus [1] reviewed the existing literature on the effect of using gadgets on children. The review asserts that modern technology helps children to develop learning abilities, competitive skills, and other educational skills faster. However, prolonged exposure to these contemporary gadgets will negatively affect children's mental and motor abilities. The work recommends further research on how technology and media affect children's development.

Radesky et al. [37] reviewed the literature on increased screen time and its effects on the behavior and development of children from birth to eight years of age. Various behavioral problems, such as aggressive behavior, attention problems, and health problems, including sleep and obesity, are associated with technology usage in early childhood.

*4.1. Cognitive*

Suzy Tomopoulos et al. [11] have researched media exposure in infant and child development. The study objective was to know if long-term exposure to media and its content in six-month-old babies would influence their development at 14 months. It was a longitudinal study of 259 participants involving mothers (from low socioeconomic backgrounds) and infants at Bellevue Hospital Centre in New York, NY, USA. From November 2005 to January 2008, their research focused on early infancy development. The data was collected through questionnaires and by interviewing mothers. Infants were exposed to electronic media (TV, video games, movies, videos) at home for different periods. Out of 259 babies, ten were not exposed to any media. The information was collected from mothers about their baby's participation in the media during 24 h. The two variables of the study were the overall time of exposure in minutes for an infant and the content of exposure in minutes. The study focused on three media content areas: educational, non-educational, and adult-oriented. Educational content consisted of child-centered educational animated shows (Sesame Street and Blue's Clues) for children aged 2 to 6 years. Non-educational content comprised of kids' cartoons (SpongeBob SquarePants, for example) without involving much violence. News, games, and talk shows were included in adult-centered content. The results showed that 15.8% (41 infants) had media exposure of fewer than 60 min, and 3.9% (10 infants) had no media exposure. Multiple linear regression analyses demonstrated that the length of media exposure during the first six months of life correlated with lower cognitive and language development at 14 months old.

Additionally, the exposure to adult-oriented content in 6-month-old infants negatively impacted the developmental process at 14 months. Overall prolonged exposure to technology and media content was related to lower cognitive and language development during early infancy. The result of this long-term study strongly suggests that children under two years of age should not be exposed to any electronic media and technology.

In an exploratory study, Caroline Ahearne et al. [12] examined touchscreen technology device usage in 82 children (47 males) aged 12 months to three years. In 2014, the study was organized at a university hospital in Ireland for five months (May to September). Parental questionnaires were part of the study. The questionnaire analyzed the children's access to touchscreen devices, duration of use per day, ability to unlock the screen, perform the swiping actions, and recognize and interact with certain touchscreen technology features. The results revealed that 87% (67 out of 82) parents owned a touchscreen device, and 87% (58 out of 67) parents offered technology devices to their kids for about 15 min each day. Out of 58 parents, 53 stated that their infant could perform swiping actions on the screen, 29 reported that the child could unlock the device, 36 parents had specific applications downloaded for the kid, and 33% (19/58) toddlers could perform all types of interactions with the touchscreen technologies in between 24 to 36 months of age. The

researchers determined that infants are accustomed to touchscreen technologies before their first birthdays, and a two-year-old toddler can identify and interact with touchscreen devices. Children acquire skills to handle touchscreen technologies in the early stages of their infancy.

　　Shoshana Dayanim and Laura L. Namy [38] studied the impact of educational video viewing on infants to learn baby signs (symbolic gestures). It was longitudinal research of 92 infants, aged 15 months old, conducted for about four weeks in greater Atlanta, GA. Researchers used 18 target sign objects (airplane, apple, dog, flower, for example). The participants were divided into four groups: 1. Video viewing alone (20 infants), 2. Co-viewing with a parent (27 infants), 3. Parental teaching (21 infants), and 4. Control (24 infants). Seven babies were dropped from the study by the second week. Children in the video-watching groups were given a DVD that contained videos instructing baby signs to infants. The babies in all three experimental groups were subjected to sign learning four days a week for about 15–20 min a day for up to 3 weeks. Video-alone group parents were told not to interact with their kids while watching. At the same time, the co-viewing group parents were instructed to keep their children engaged by directing the child's focus toward the television, repeating and emulating signs. In the parental teaching group, the parents taught signs (gestures) to their infants with the help of a picture book. Control group infants had no exposure to any signs. All the groups were instructed not to use or imitate signs beyond the allocated education sessions. After three weeks of instruction sessions, kids were not exposed to any signs for the next week. In weekly learning assessments, parents were given a vocabulary checklist to check if the infant understood all 18 items in the target sign list. The experimenter of the study designed a laboratory-based forced-choice task to evaluate each infant's sign apprehension by placing two different target objects photographs and then making one of the signs to assess if the kid could comprehend it. Finally, on the fourth-week visit, the infants performed an elicited sign-producing task. To prompt the production of signs, the researcher displayed photographs of the target objects individually and asked the infant, "Can you say this sign with your hands?" or "Can you demonstrate this with your hands?" and expected to produce the sign with their hands. Thus, researchers measured a child's learning ability and retention skills. The ANOVA results of the study of all four groups for every lab visit showed that there was no effect on one-week exposure, but there was positive effect after week two. During week three, in all three experimental groups, parents reported evidence of positive learning effect after the second week. Tamhane's post hoc analysis showed essential distinctions between the three experimental groups and the control group. Tukey's HSD (Honestly Significant Difference) test demonstrates that co-viewing group parents reported that their children generated considerably more signs than infants of the video-alone group. The study demonstrated that videos could help babies learn signs with or without their parents' support. Thus, infants can learn and obtain information through educational video watching.

*4.2. Language*

　　McClure et al. [5] analyzed "why video communication is considered an exception to media restriction for infants and toddlers". Researchers conducted a virtual media usage survey of 183 parents with infants between six and 24-months old in the Washington, D.C. metro area. The assessment included questions about general parental information, infant development data, and the baby's age, media usage, and exposure to video chats the day before the survey, along with the best two explanations for using video chat with their child. The results indicated that 60% of parents use video chats a few times a month, and 37% of the participants reported that they extensively use video communication once a week. Video call use was high among 84% of babies (6–16 months old) and 88% of older infants (17–24 months old). The participants used FaceTime (48%) and Skype (41%) for video calling. Communicating with remote family and socializing were the major motivating reasons for video chats. The findings also suggest that based on the survey results, the researchers divided the infants into two distinct media user groups: high-usage

and low-usage. However, children who did not use any other technology devices used video calls. The study concluded that parents expose infants to media via video talks at an early age. Communication through video calls has been perceived positively by parents of children under two years. Video chats are considered an exception to other media usage, as they aid infants and toddlers in communicating with distant families and preserving social bonds. Further research is vital to understand precisely how infants participate socially and emotionally when communicating during video conversations and to understand the developmental effects on babies if exposed for longer durations. Supplementary research is crucial to understanding when babies can be exposed to technology and video communication by considering the AAP's recommendations.

Duch et al. [39] investigated the association between technology usage and language development in Hispanic children. This longitudinal and cross-sectional study comprised 119 toddlers (38.1% boys) and caretakers (primarily mothers) who were part of an urban early head start (EHS) program for 9–12 months. The children in the sample were between one month old and 5.5 years. Demographic information and data about the infant's playtime, screen time, and family habits during leisure time were collected from parents through questionnaires. The toddler's language development was measured by the Ages and Stages Questionnaire: a parent-completed child monitoring system, Third Edition (ASQ3). The questionnaire contained questions related to various developmental areas, such as communication, motor skills, problem-solving, social interaction, and development, and the study consisted of a caretaker's interview. In addition, the information about the screen time of parents, child, and child-parent together (co-viewing) in the previous 24 h and the type of technology used to access media, such as TV, mobile phones, DVD, or computers, was collected. Screen time was categorized as child-directed (list of children shows) and adult-directed (adult shows watched by adults), and screen time was measured in the duration of below or above two hours. Various statistical and bivariate analyses showed the link between kids' screen time and development. The study results revealed that a toddler spent 2.09 h watching child-directed media and 1.29 h per day viewing adult-directed media, and the child's average TV time accounted for 3.29 h a day. During early infancy, babies' exposure to adult-directed media was more. In contrast, child-directed media watching was directly proportional to the child's age. Smartphones are kids' most common technology, with 33.5% watching television, and only 7.9% using computers. The average time of background TV turned on in-home was 4.46 h per day, 13.7% of toddlers and 12.5% of infants used mobile phones for about 31 min to 2 h a day, and 71% of parents stated that their child watched baby shows on DVDs. About 84% of households think that watching educational shows has a beneficial effect on infants and improves their learning skills.

Additionally, 46% of mothers watch television while breastfeeding their babies, and 38% of kids watch television while eating. Cross-sectional analysis results show that babies and toddlers who watched TV beyond 2 h a day were linked with 5.5 times lower communication scores than toddlers watching less than 2 h of TV. The long-term analysis demonstrates that longer exposure to television was correlated with less communication skill development. Furthermore, exposure to child-directed media is positively related to less communication than adult-directed media content. In conclusion, this research revealed the adverse effects of technology usage on a child's language development abilities.

### 4.3. Physical

Cespedes et al. [40] analyzed long-term relationships between television viewing and sleep time from early childhood to mid-childhood. The longitudinal study comprised 1864 mothers and children. The participants of this study (born between 1999 and 2003) were the participants in Project Viva conducted at Harvard Vanguard Medical Associates practices in Massachusetts. The researchers conducted one-on-one study visits with mother and child during infancy, early childhood, and mid-childhood. Mothers completed questionnaires annually for six years. Demographic data such as parents' education, income,

ethnicity, and kid's average sleep timing and TV-viewing behavior were collected through the questionnaires from infancy (six months) through mid-childhood (7 years). Firstly, the details of the TV watching timings were collected.

Mothers of six- to 12-month-old infants reported the average time infants spent with the background TV on and where babies could see the TV. In each in-person assessment from two to seven years, mothers separately described the child's average TV and video viewing hours during weekdays and weekends. At four to seven years of assessments, information regarding the presence of a TV in children's bedrooms was also gathered. Secondly, detailed data about the sleep time of children was collected. At six months, average sleep time during the late morning and afternoon naps and night sleep of the prior month were reported by mothers. From two to seven years, mothers reported the past month's average sleep duration on weekdays and weekends. The researchers used regression models to assess the longitudinal correlations between a child's television exposure and sleep duration. The results of the study revealed that at six months, an infant's sleep time is about 12.2 h per day; at age one, sleep duration increased to 12.8 h a day; and sleep time remained constant from two to five years, whereas at seven years of age, the duration of sleep decreased to 9.8 h per day. Television viewing behavior surged between six months and four years from 0.9 to 1.7 h a day.

In contrast, at seven years of age, children's TV time decreased slightly to 1.6 h. At age four, 15% of children had a TV in their bedroom, which increased to 23% by age seven. Ethnicity and race had a significant impact on access to TV in bedrooms. Various cross-sectional models showed that more TV watching was linked with less duration of slumber time. Multivariate adjustment indicated that every extra hour of watching TV was correlated with three fewer minutes of sleep per day in six-month-old babies, two fewer mins a day in three-year-olds, and six fewer mins per day in seven-year-old children. Having a TV in the child's bedroom between four and seven years of age was linked with 32 min of less sleep time a day. Moreover, the relation between TV watching and decreased bedtime sleep was stronger among boys. The study concludes that watching television and having a bedroom TV adversely impacts the onset of sleep from early infancy to mid-childhood.

According to the American Academy of Sleep Medicine, the recommended amount of sleep for children according to age is presented in Table 3.

**Table 3.** The recommended total amount of sleep for different age groups of children [41].

| Age Group | Age | Recommended Sleep (Hours/Day) |
|---|---|---|
| Infants | Younger than 12 months | 12 to 16 h |
| Toddlers | 1 to 3 years | 11 to 14 h |
| Preschoolers | 3 to 5 years | 10 to 13 h |
| Children | 6 to 12 years | 9 to 12 h |
| Teenagers | 13 to 18 years | 8 to 10 h |

In a cross-sectional study, Cox et al. [42] examined the relationship between television viewing habits, TV content, food consumption, physical activity, and the body mass index (BMI) in preschool-aged children. A survey was conducted for 135 preschool kids, aged between two and six years, and their mothers, who resided in Melbourne, Australia, between June and September 2010. In the study sample 60% of children were girls, and 14.8% of children were obese. The data were collected through questionnaires from mothers. The questionnaire included questions about the child's demographic information, parental birth details, and other socioeconomic (SES) details. Mothers were asked to keep a diary to note the time spent by the child watching TV, DVDs, and other videos for three successive days. The survey collected data about the kids' screen time, content type and title of the show, TV channel, food/drink intake while watching television, and type of food (drinks, fruits, veggies, fast foods, dairy products.). Television programs were categorized into commercial (TV shows that include in-program commercials) and non-commercial (without in-program

commercials) TV watching. The child's total energy intake of food/drink while watching TV was calculated in kilojoules. Information about kids' physical activities was evaluated using a physical activity questionnaire for preschool-aged children (pre-PAQ). Parents described the type of activity and the time spent by their child on physical activity the previous day and prior weekend. Additionally, information regarding the kids' diets was assessed with the validated Children's Eating and Physical Activity Questionnaire (EPAQ) to measure the consumption of energy-rich foods and sugary drinks intake. Mothers reported the kids' previous day's intake of 10 foodstuff and beverage categories to measure how frequently the child had obesogenic foods.

The results of the survey stated that there was a positive association between kids' BMI z-scores and weekday TV watching. Kids allocated 90 min more per day to watching non-commercial programs. Watching television, consuming poorly nutritious food, and regular intake of fast foods are positively correlated. Substantial relationships were noticed between children's weekday and weekend television-watching behavior and the time spent without physical activity. Thus, it shows a significant association between viewing TV and a child's BMI z-score. The study concludes that watching television may cause obesity in preschool-aged children due to inactive/sedentary activities and consuming food while watching.

Amy I. Nathanson [43] reviewed the literature on the impact of technology on sleep in young children. The review focuses on existing research that examines the effects of technology use during early childhood under six years of age and various sleep outcomes in children associated with technology. Nathanson stated some primary considerations in his literature review. This article also gives recommendations for future work and suggests the need for further longitudinal research to comprehend the association between sleep and technology usage in younger kids. Lastly, the author gives some guidelines to clinicians regarding how to evaluate and tackle technology-related sleep issues.

### 4.4. Social and Emotional

Sarah M. Coyne et al. [44] studied the relationship between temperament, media emotion management, and problematic media use in toddlers. The study was conducted in 2019 and comprised 269 children (2–3 years) (132 males, 115 females, with 3 unaccounted) and their primary caregivers in Denver, Colorado. Questionnaires for caretakers and observational assignments to study child's emotional reactions were part of the study. The Early Childhood Behavior Questionnaire short form [45] was used to measure the child's personality and behavior. The questionnaire summarized questions on how frequently caretakers would provide media devices to children to manage their emotions to evaluate media emotion regulation. Toddlers' problematic media usage data was collected using the Problematic Media Use short form [46], which included questions relating to fixation with media, disinterest in other activities, patience, and serious complications due to media use. Observational tasks were designed to examine the kids' emotional reactions while viewing a well-known television clip (from Daniel Tiger's Neighborhood). The researchers applied a Multivariate Analysis of variance and found a relationship between emotion management and problematic media usage in toddlers.

Moreover, problematic media use was directly correlated with media usage time and inversely correlated with effortful control. Parents expose their kids to technology to normalize their extreme emotional behaviors. Such kids demonstrated more challenging conduct and tantrums when the media was confiscated. Researchers concluded that parents should not be using technological devices to control or distract the infant's complex emotions. Doing so would hinder the developmental process of regulating emotions in toddlers and young children.

Radesky et al. [6] conducted an observational survey to examine the correlation between children's socio-emotional development and mobile technology use. The study comprised 144 parents and children (15–36 months). The researchers accumulated participants' demographic data. The survey included questions to parents enquiring about the

likelihood of permitting digital technology usage by children in different scenarios, such as to calm the kids while eating, accompanying parents in public places, to complete their daily chores, or during sleep times. The survey also included questions from the parents' opinion survey to evaluate the effect of parental control on children's development and social and emotional behavior. The multivariable regression analysis revealed that 40.3% (58 of 144) of children were offered mobile technology as a comforting tool when upset. It can be deduced that a strong relationship exists between toddlers' emotional dysregulation and technology use. Low-income parents who had less control over a child's behavior were inclined to provide technology as a means to calm their kids or relax them. This research suggests a further longitudinal study to comprehend the associations between technology usage and childhood development.

In a longitudinal study, Radesky et al. [47] evaluated the association between self-regulation problems (such as difficulty with emotional regulation, self-soothing sleep, and attention) in infants during early infancy and exposure to media (TV and video watching). The study involved 7450 children, and the data was collected from the parents during infants' nine months and 24 months (as their children grew from nine to 24 months). The information was gathered through interviews, questionnaires, and developmental assessments. During the assessment of kids at nine months and at two years, the parents were asked to complete a modified Infant Toddler Symptom Checklist (ITSC) to assess the regulatory behavior and problems in children. The checklist included questions about the child's sleep issues, feeding problems, fussiness, handling emotions, and behavior. Parents were asked to independently describe their kids' screen time on weekdays and weekends when they were two years of age. The child's media exposure hours were measured. The children's and parental demographic information, including race, gender, age, weight, parents' age, socioeconomic status, languages spoken at home, and maternal details, were collected. The various bivariate regression analyses showed that, on average, a two-year-old child's media exposure was 2.3 h a day. African American children in low-income families with a single parent, as well as children with a mother suffering post-partum depression, were more likely to be considered as having poor self-regulation. Babies with poor self-regulation at nine months had 0.23 h more media exposure than infants with moderate or no self-regulation problems. Children with persisting dysregulation problems at nine months and two years spent more hours in front of media devices and had screen time exceedingly more than 2 h a day at age two. The researchers concluded that early childhood self-regulation problems correlate with high media use. Thus, this strongly connects child emotional dysregulation and media exposure.

In Table 4, we have tabulated some research studies on the effects of media usage and technology on child development between 2010 and 2021. It provides a concise summary of each study's key findings and sources. Thus, it gives an overview of each study and the characteristics of the study in brief, emphasizing the importance of media and technology in shaping child development.

**Table 4.** Attributes of studies included in the research.

| Year | Country | Research Methods | Age (years) | Sample Size | Findings | Reference |
|------|---------|------------------|-------------|-------------|----------|-----------|
| 2010 | US | Questionnaires, Maternal interviews | 0.5–1.2 | 259 | More prolonged exposure to media affects cognitive and language development. | [11] |
| 2012 | Australia | Parental questionnaire | 2–6 | 135 | Sedentary activities and diet contribute to childhood obesity. | [42] |
| 2013 | N/A | Questionnaires and interviews | 0.5–5 | 119 | Media usage affects language development abilities. | [39] |
| 2014 | US | Maternal Questionnaire | 1–7 | 1864 | Sleep problems in children. | [40] |

**Table 4.** *Cont.*

| Year | Country | Research Methods | Age (years) | Sample Size | Findings | Reference |
|---|---|---|---|---|---|---|
| 2014 | US | Interview, Questionnaire, and developmental assessments | 0–2 | 7450 | Self-regulation problems are linked to excessive media usage. | [47] |
| 2015 | US | Laboratory-based observational tasks and | 1.2 | 92 | Babies learn and acquire knowledge through educational content. | [38] |
| 2015 | US | Online survey | 0.5–2 | 183 | Video chats are considered an exception to other media usage. | [5] |
| 2016 | Ireland | Parental questionnaire | 1–3 | 82 | Infants are learning to handle touchscreen technologies. | [12] |
| 2016 | US | Observational survey | 1–3 | 144 | Emotional and behavioral problems. | [6] |
| 2016 | US | Literature review | 0–8 | - | Media benefits and effects on child health and behavior. | [37] |
| 2018 | Turkey | Literature review | 0–7 | - | Developmental risks due to long-term smart device usage. | [14] |
| 2018 | Pakistan | Literature review | 0–7 | - | Excessive modern device use impacts a child's cognitive and motor skills. | [1] |
| 2019 | US | Online surveys, Questionnaires | 0–3 | 326 | Relationship between family media ecology and child's media usage. | [32] |
| 2021 | US | Questionnaires, Observational tasks | 2–3 | 269 | Prolonged use of technology devices causes emotional regulation issues. | [44] |
| 2021 | US | Literature review | 0–6 | - | Impacts of technology on sleep during early childhood. | [43] |

Table 5 provides an overview of the impact of technology on various child development domains across different age groups. In this table, we have grouped the studies according to childhood developmental domains, such as cognitive, language, physical, and socio-emotional. The study outcomes range from positive impacts, such as infants learning new information and acquiring knowledge through exposure to technology, to negative impacts, such as emotional dysregulation, behavioral problems, and obesity in preschool-aged children due to inactive/sedentary activities.

**Table 5.** Impact of technology on different age groups of children.

| Development Domain | Age (Years) | Impact | Technology Type | Study Outcome | Reference |
|---|---|---|---|---|---|
| Cognitive | 1.25 | Positive | Television, smartphones, tablets, and computers. | Infants learn new information and acquire knowledge. | [37] |
| Cognitive, Physical | 1–3 | Positive | Smartphones, tablets. | Young kids acquire skills to handle touchscreen technologies. | [12] |
| Cognitive, language | 0.5–1.2 | Negative | TV, video games. | Affect cognitive and language development. | [11] |
| Language | 0.1–5.5 | Negative | TV, mobile phones, DVD, and computers. | Affect the child's language development. | [38] |
| | 0.5–2 | Positive | Video Chats on mobile devices. | Enhancing communication skills. | [5,41] |

**Table 5.** *Cont.*

| Development Domain | Age (Years) | Impact | Technology Type | Study Outcome | Reference |
|---|---|---|---|---|---|
| Physical | 2–6 | Negative | Television | Obesity in preschool-aged children due to inactive/sedentary activities. | [41] |
| | 1–7 | Negative | Television | Affects sleep onset and sleep duration from infancy to mid-childhood. | [39] |
| | 0–6 | Negative | TV, Video games, smartphones, tablets, and computers. | Impacts on child's sleep. | [42] |
| Socio-emotional development | 0–2 | Negative | TV, and other mobile devices. | Excessive screen time causes emotional dysregulation problems. | [46] |
| | 1.25–3 | Negative | Mobiles, tablets, computers, and gaming tools. | Emotional and behavioral problems. | [6] |
| | 2–3 | Negative | Television | Impacts emotional dysregulation and behavioral issues. | [43] |

## 5. Impact of COVID-19 Pandemic on Children's Screen Time

The COVID-19 pandemic has altered human lives in myriad ways. Social distancing, lockdown restrictions, and other business closures affected life worldwide [48]. The pandemic has also hampered the education system by closing schools worldwide. Thus, students were introduced to distance or remote learning at all levels of education [49]. With the school closures and being confined to homes, the children's screen time has changed significantly, resulting in more prolonged exposure to screens, impacting them negatively [50]. Several studies suggest a correlation between the COVID-19 pandemic and children's screen time. During the first wave of the COVID-19 global pandemic, children's screen time drastically increased for all age groups [51]. A survey in Ontario (Canada) found that children's screen time increased from 2.6 to 5.9 h per day after the pandemic [52]. A cohort study conducted in the U.S., consisting of 228 children between the ages of four and 12 years, revealed that the average total screen time increased by 1.75 h per day during the first wave of the pandemic and by 1.11 h per day during the second wave of the pandemic period, compared to the pre-pandemic period [22]. Another American study showed that children's screen time increased by 1.62 h during the pandemic compared to the pre-pandemic [53].

Another study reported that 79.1% of parents stated that their kids' screen time exceeded more than 2 h a day during online education [54]. A longitudinal study in Canada stated that the pandemic has significantly increased children's screen time. Children's recreational screen time use has almost doubled when compared with post- and pre-pandemic situations. Before the pandemic, the average increase in screen time was one hour per week, whereas, during the pandemic, the same children had an increase of 11 h of recreational screen time per week [55]. An observational study conducted in several European countries, which included 8395 children, examined the correlation between the screen time and physical activity of children and concluded that the COVID-19 pandemic restrictions had a negative impact on children's physical activity and screen time [56]. In countries such as Australia, Netherlands, France, Italy, Germany, the Republic of Korea, the U.S., Spain, the UK, and China during the lockdown, children from three to 17 years of age spent long hours in front of digital screens. The COVID-19 pandemic impacted even younger children in the early developmental stages. Screen time in toddlers during the lockdown increased drastically compared to pre-lockdown rates. During the pandemic,

children's exposure to screens worsened in different countries with longer duration of lockdowns [51].

Increased media, excessive digital device use, and limited or fewer outdoor activities during the pandemic may have led to long-term negative impacts on children's health, leading to the early onset of various eye problems such as myopia [57]. A study in Turkey has examined the relationship between parenting practices and child screen time during COVID-19. The study's results showed that nearly 71.7% of parents reported that their child's screen time increased by an average of six hours a day during the pandemic. Age, gender, household income, screen time rules, and inconsistent parenting practices were significant predictors of child screen time during the pandemic [58].

Overall, there is growing evidence that the COVID-19 pandemic and distance education drastically impact children's screen time worldwide. Figure 5 illustrates the connections among COVID-19 and technology use in children [59,60]. Prolonged screen time use in children can affect their physical, cognitive, and emotional well-being and have adverse developmental impacts [61].

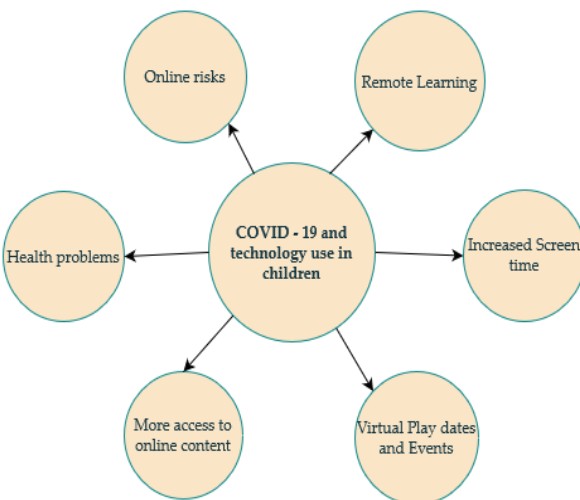

**Figure 5.** COVID-19 and technology use in children.

## 6. How Can Parents Minimize Screen Time

Technology has become an integral part of children's daily lives. In today's world, kids are exposed to various technological devices for various purposes, such as education, entertainment, and communication. Technology use by young children has several potential negative impacts on their development, including decreasing playtime, increasing time spent in front of screens, and increasing the amount of time spent alone. Thus, parents and providers need to be aware of the potential adverse effects of technology on young children, so that they can make informed decisions about the appropriate technology use for their children [14]. Adults' screen time usage influences the child's screen time behavior [62]. According to the AAP recommendations, children under two to five years of age should not exceed one hour of media screen time. However, recent research [63,64] demonstrates that 80% of two-year-olds and 90% of three-year-olds did not meet AAP's recommended guidelines [65,66]. The factors correlated with the exceeding media usage guidelines were mothers' screen time use and children being confined to homes where access to digital screens is higher [51,64].

*Control Measures for Parents to Limit Child's Screen Time*

For parents, it is essential to establish control measures for a child's screen time. Here are some measures that parents can implement to limit their child's screen time [54,62,67–69]:

1.  Carefully select the media content for the child to view.
2.  Adopt co-viewing and participate in content discussions with the child.

3.  Teaching kids the strategies of viewing skills, such as interpreting the program content and carefully observing details.
4.  Limit the child's media watching time, content, and type.
5.  Discourage media use as a comforting tool.
6.  Encourage digital detox time and engage in other activities away from screens.
7.  Set up children's bedrooms free from TV and other electronic media devices.
8.  Following the 20-20-20 rule, i.e., for every 20 min of screen use, take a break and look at something 20 feet away for 20 s. Practicing the 20-20-20 rule will help alleviate digital visual strain and safeguard the eyes from various vision disorders.
9.  Parents should set lower screen-time limits for themselves.
10. Parents should be wise in replacing screen time with other activities, such as outdoor playtime, skill development, sleep, interaction with friends and family, studies, and physical activities vital for children's overall health and development.
11. Parents should be educated on the consequences and possible detrimental effects of excessive media and screen usage.

## 7. Specific Recommendations

### 7.1. The American Academy of Pediatrics (AAP)

AAP recommends the following media usage guidelines based on children's development stage [70]:

1.  Children under two should not be exposed to media, excluding video chatting.
2.  Toddlers and preschoolers should have at most one hour a day of high-quality programming.
3.  Teenagers should avoid allowing the media to replace significant activities, such as physical exercise, sleep, quality time with family, and media downtime.
4.  Media users of all ages should act as media mentors. In addition, parents are encouraged to co-view with their children.

Table 6 specifies the AAP recommendations for media usage according to the age group of children. These recommendations provide various advantages for children. Excessive media use can negatively impact a child's physical and mental health, including sleep problems, obesity, eye problems, and other behavioral issues. Following the above guidelines helps children maintain beneficial media use and cultivate healthy habits, such as quality sleep, social engagement, outdoor play, and physical activity. Parents co-viewing with children and encouraging children to watch educational content can foster positive media usage. These guidelines suggest clear and concise advice for parents by educating them about healthy media habits. Parents must follow procedures to balance technology's advantages and possible harms and protect their infant's health. Altogether, AAP's media usage recommendations for children can nurture healthy media habits and safeguard against adverse impacts, improving children's development and overall well-being.

**Table 6.** The American Academy of Pediatrics (AAP) guidelines for children's media usage [71].

| Age | Recommendations |
| --- | --- |
| Babies below 18 months | No screen media except video chats |
| Children between 18–24 months | High-quality programming content and co-viewing with a parent |
| Children between 2–5 years | One hour per day and high-quality programming content |
| Children 6 years and above | Limited media screen time and limitations on viewing different media types |

The graph in Figure 6 allows us to visualize and analyze the relationship between age and recommended screen time, with the x-axis representing the age of children (in years) and the y-axis representing the recommended screen time (in minutes).

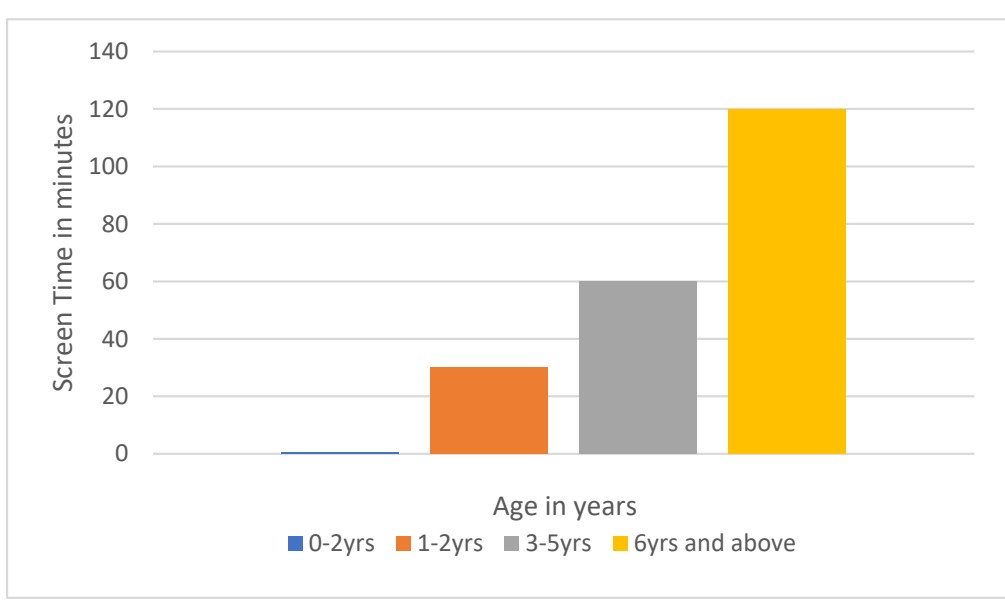

**Figure 6.** Recommended screen time for children of different age groups in minutes.

7.1.1. Recommendations for Pediatricians

1. Encourage asking the parents about family media usage time, child's media use behaviors, and locations of media use.
2. Helping families develop a Family Media Use Plan using a website such as www. healthychildren.org/MediaUsePlan, accessed on January 25, 2023.
3. Educating parents about children's early brain development and the importance of cognitive, language, and socio-emotional skill development.
4. Advising parents to discourage using media screens except for video chats for children under 18 months and not allowing the children to use media on their own.
5. Encouraging parents to select high-quality program content for their children.
6. Recommending no screen use during mealtimes and before bedtime.

7.1.2. Recommendations for Families

1. No digital media usage for kids younger than 24 months.
2. Co-view with children and helping them understand and learn what they are watching.
3. Turn off TVs and other digital devices when not in use.
4. Do not use media as a calming tool for a child.
5. Monitor the child's media content and media habits.
6. Turn on the "do not disturb" option during their playtime with children, mealtimes, and when making them sleep.

*7.2. World Health Organization (WHO)*

The World Health Organization (WHO) has released new guidelines for how much children should spend on screen. Table 7 tabulates these guidelines that recommend children under two years should not have screen time and children aged two to five should not have more than one hour per day. The guidelines are the first set of formal recommendations from the WHO on screen time use for children, and they encourage parents to set limits on how much screen time their children can have [72].

*7.3. IEEE Standards Association*

To better design digital products and services considering children, the IEEE Standards Association has published "IEEE 2089-2021, IEEE Standard for an Age-Appropriate Digital Services Framework" based on the 5 Rights Principles for Children. This framework offers practical steps that product development teams, suppliers, and process evaluators can

follow to ensure their online services and products are safe for children. This standard is intended to help organizations design their services by considering children [73].

**Table 7.** WHO guidelines on sedentary screentime behavior for children below 5 years [72].

| Age | Recommended Screen Time per Day in Minutes |
|---|---|
| Infants under 1 year | 0 min |
| Children 1–2 years | No more than 60 min |
| Children 3–4 years | No more than 60 min |

## 8. Discussion

Technology's usage has both positive and negative effects on young children. It can offer educational benefits, such as helping toddlers learn new skills, improving hand-eye coordination, and stimulating curiosity and creativity.

It can also provide entertainment and help parents keep their children occupied while they attend to other tasks. Figure 7 shows some of technology's positive and negative impacts on children.

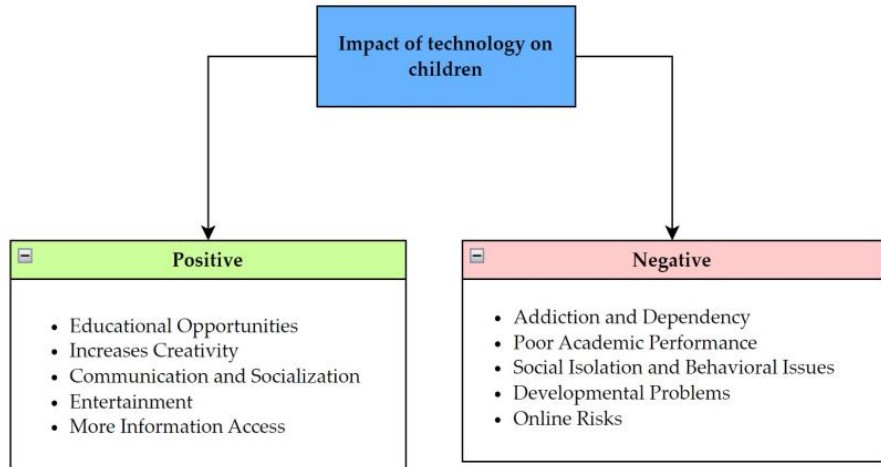

**Figure 7.** Positive and negative impacts of technology use in children.

On the other hand, excessive use of technology can negatively affect toddlers, including delayed language development, poor social skills, and behavioral problems. It may also lead to addiction, poor sleep quality, and obesity in children. We also provide information (Table 8) on how different media types impact a child's developmental domains in positive or negative ways.

**Table 8.** Impact of different media types on child development domains.

| Media Type | Impact on Developmental Domain | | | | Level of Impact | | |
|---|---|---|---|---|---|---|---|
| | Cognitive | Language | Physical | Socio-Emotional | 0–2 years | 2–5 years | 5–7 years |
| Television | +/− | +/− | − | +/− | Low | High | High |
| Smart Phone | +/− | +/− | − | +/− | Low | High | High |
| Video Games | +/− | +/− | +/− | +/− | Low | Low | High |
| Educational Apps | + | + | Neutral | + | Low | Low | High |
| Social Media | − | +/− | Neutral | − | Low | Low | Low |
| Mixed Reality | + | + | + | + | Low | Low | Low |

The casual loop diagram in Figure 8 describes the key variables and nodes illustrating the positive and negative relationships between a child's developmental domains and various types of media.

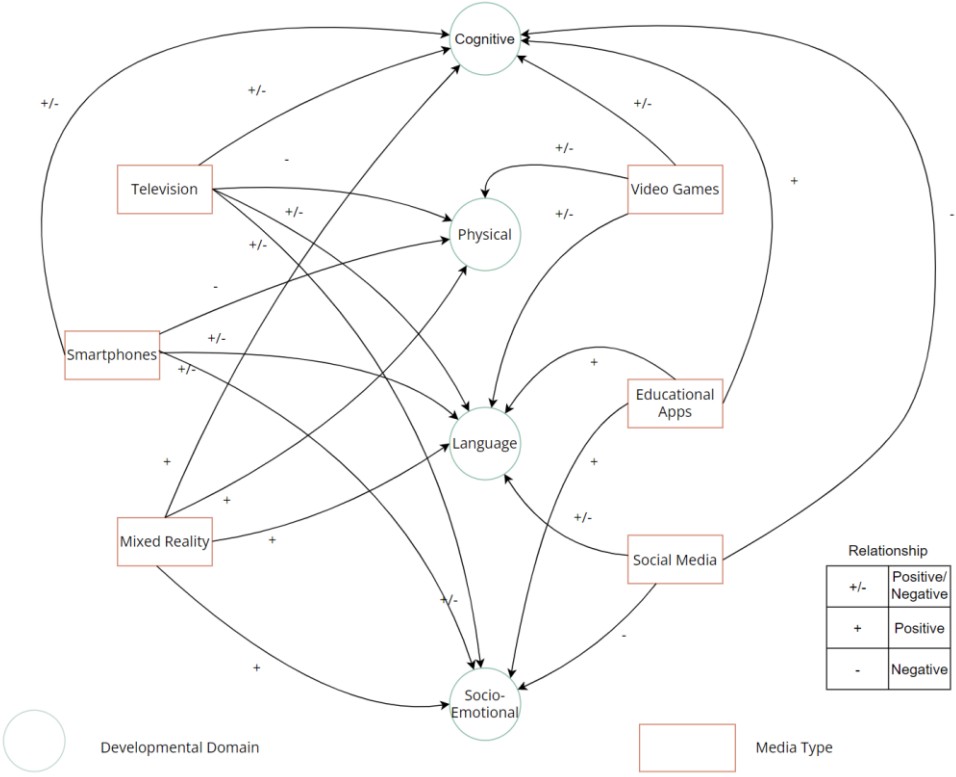

**Figure 8.** Casual Loop Diagram of the impact of different types of media on child development domains.

Therefore, parents should strive to balance using technology as a tool for learning and entertainment while prioritizing other forms of knowledge and playtime for their children. Supervising technology usage and ensuring that children engage in age-appropriate content in moderation is essential.

**Recommendations for Improving Child Developmental Domains**

Improving child developmental domains is critical for enhancing a child's overall development. Cognitive development involves problem-solving, reasoning, and attention. Physical development in children refers to the growth and changes in a child's body and motor skills, nutrition, sleep, and overall health. Language development involves speech and communication skills. Social and emotional development involves relationships, self-regulation, and emotional expression. Following are some of the recommendations for parents to improve various child development domains:

**Cognitive:**

1. Encourage play such as pretend play, physical play, and construction play. Play is fundamental for a child's development across all development domains.
2. Engage children in puzzles and games, such as jigsaw puzzles, matching, and memory games. This can improve their memory and problem-solving abilities.
3. Provide real-world examples while explaining ideas and concepts help in understating the world and knowledge development in children.
4. Encourage hands-on learning experiences such as gardening, cooking, conducting science experiments and other practical experiences, provide opportunities for a child to explore different areas and topics and help them to develop critical thinking and reasoning.

5.  Encourage Practicing mindfulness and meditation, healthy eating habits, and good sleep habits help promote children's cognitive development.

**Socio-emotional:**

1.  Encouraging face-to-face interactions with family members, friends, and peers.
2.  Encouraging self-expression by encouraging children to express themselves through music, art & crafts, dance, or writing.
3.  Limiting screen time as long exposure to screens hinders social and emotional development.
4.  Practicing empathy by encouraging the kids to consider others' perspectives and feelings.
5.  Engaging in role-playing, which can be a fun and an effective way to help children practice empathy and social communication skills.
6.  Fostering positive relationships with family and others. This can help them develop strong social connections and improve emotional intelligence.

**Language:**

1.  Reading aloud to children is one of the best ways to help them develop their vocabulary and language skills.
2.  Engaging in regular conversations with children, encouraging self-talk, and discussing their feelings and interests will improve their conversational abilities.
3.  Playing word games helps the kids expand their vocabulary skills.
4.  Using descriptive language and singing songs to children helps develop speech and language as they learn new words, sentences, and concepts.
5.  Encouraging children to write letters and short stories to use their creativity and enhance their writing skills.

**Physical:**

1.  Promoting engagement in physical activities, such as playing sports, walking, biking, running, and jumping to strengthen muscles and bones.
2.  Encouraging kids to play outside can provide various health benefits, improve creative play, and explore their surrounding nature.
3.  Reducing sedentary activities, such as sitting for long durations watching TV or playing video games.
4.  Promoting healthy eating habits by providing the child with balanced healthy meals.
5.  Cultivating good sleep habits by establishing regular sleep schedules helps improve a child's physical and mental health.

Figure 9 tabulates some of the recommendations for improving cognitive functions [74–76], social and emotional development in children [77], language development [78], and physical development in children [79,80].

Improving child developmental domains is essential for helping children become healthy, happy, and successful adults. Parents, educators, and caregivers must provide children with the necessary support and resources to help them develop in all areas of their lives.

**Positive and Negative Implications of Artificial Intelligence (AI) on Children's Screen Time**

AI can have positive and negative implications on kids' screen time [81,82].

**Positives of AI:**

1.  AI can offer many educational opportunities and resources to enhance a child's learning experiences and improve academic skills.
2.  Children can use generative AI to explore new ideas and concepts and enhance creativity by co-viewing with parents. For example, storytelling, math discussions, science query, and knowing world history using ChatGPT. Figure 10 shows some examples of interactions with ChatGPT.
3.  AI can provide personalized learning experiences and feedback to each child's preferences and needs, improving critical thinking skills. For example, IXL is an online

learning platform that offers personalized learning experiences to students in various subjects, including Math, English Language Arts, Science, and social studies.

4. AI can improve productivity by saving time for other tasks and helps in efficient time management in children. In addition, AI can help in problem-solving by breaking down complex problems into small subsets in an efficient manner, thus saving time–for example, solving difficult math problems.

**Cognitive:**

- Encourage play.
- Engage in games and puzzles.
- Using real-world examples.
- Provide exploration and practical experiences.
- Practice healthy routines such as mindfulness, a healthy diet, and sound sleep.

**Socio-emotional:**

- Encourage face-to-face interactions.
- Promote self-expression.
- Limit screen time.
- Practice empathy.
- Foster positive relationships.

**Language:**

- Read aloud to children.
- Encourage regular conversations.
- Playing word games.
- Sing songs and use descriptive language.
- Practice writing skills.

**Physical:**

- Encourage physical activity.
- Encourage Outside play.
- Reduce sedentary activities.
- Promote healthy eating habits.
- Encourage good sleep habits.

**Figure 9.** Recommendations for improving child development domains.

**Some Examples of Interactions with ChatGPT:**

VI    Factors influencing global warming for a five year old. can you make up a brief story

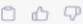

Once upon a time, there was a planet called Earth. Earth was very special because it had everything we needed to live: air, water, land, and lots of different plants and animals.

But there was a problem. People who lived on Earth started to do things that were hurting the planet. They were using cars that made lots of pollution, cutting down too many trees, and using lots of electricity that came from burning things that made more pollution.

All this pollution made the air around Earth get warmer and warmer, like when you wear too many clothes on a hot day. This is called global warming, and it's not good for the planet or the animals that live on it.

So, we need to do things to stop global warming, like using less energy and planting more trees. That way, Earth can stay healthy and happy for a long, long time!

**Figure 10.** *Cont.*

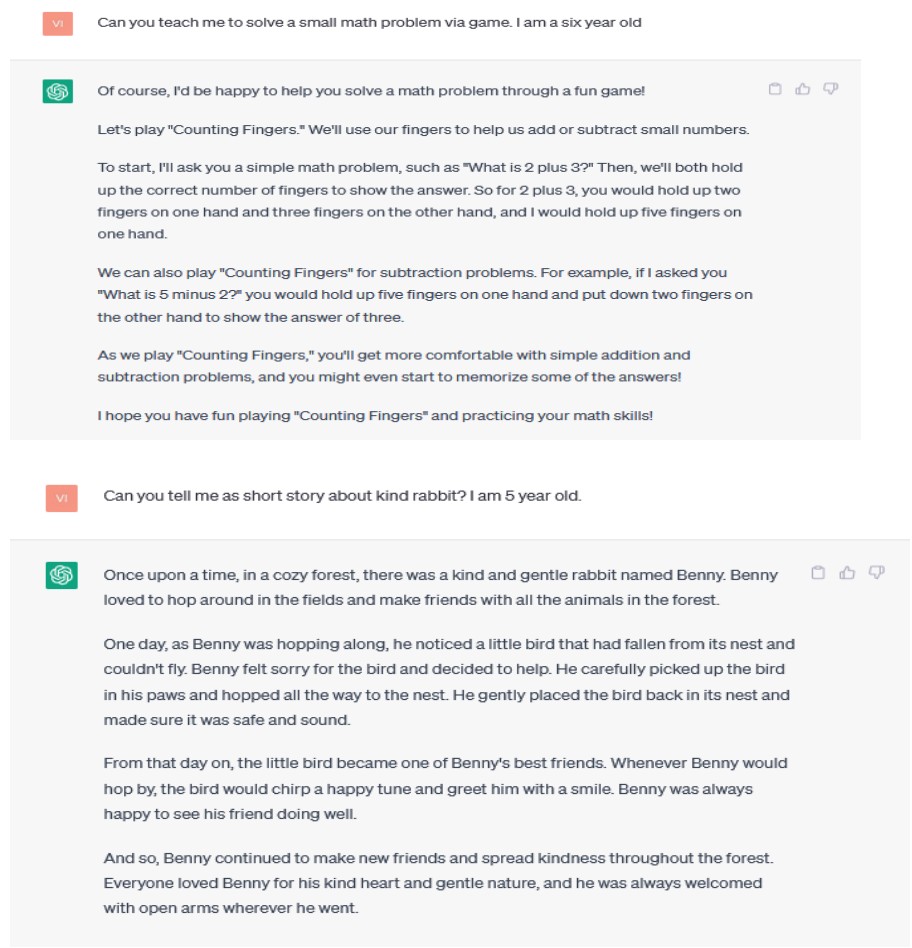

**Figure 10.** Examples of ChatGPT Interactions.

**Negatives of AI:**

1. AI can influence children's screen time leading to addiction, which can negatively impact the child's health.
2. AI discourages children's face-to-face interaction, affecting their social life and well-being.
3. AI can create dependency on technology and may discourage kids' problem-solving capabilities.
4. AI might encourage excessive screen time that might pose potential cyber risks for children, such as exposure to inappropriate content, cyberbullying, and other online risks.

Below, Figure 11 explains both positive and negative implications of AI:

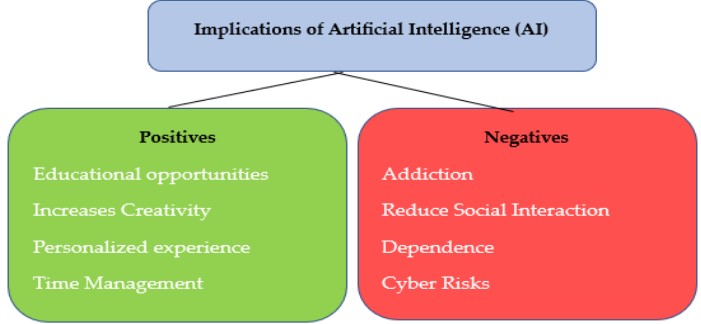

**Figure 11.** Implications of AI on Children.

#### Screen Time and Cybersecurity Challenges for Children

Children today spend long hours in front of screens for education, entertainment, and other reasons. With excessive screen time, children are more prone to potential cyber risks. The following are some significant cyber challenges for children [83]:

1.  Cyberbullying: Children who spend a lot of time on social media and other online platforms risk being bullied, harassed, or intimidated by peers or strangers.
2.  Exposure to inappropriate content: Children can be exposed to inappropriate content, such as violent or sexual material while browsing online.
3.  Addiction to screens: Excessive screen time can lead to addiction, negatively impacting children's physical and mental health.
4.  Online predators: Children who spend much time online can become targets for online predators, who may try to groom them for abuse, exploitation, or blackmail.
5.  Privacy and other security risks: Children's personal information and privacy can be at risk online, particularly if they share sensitive information or interact with strangers.

#### Online Safety Guidelines For Parents To Protect Their Children From Cyber Risks:

It is important to prioritize the safety of children online. Here are some safety guidelines to help mitigate the cyber risks to children [84]:

1.  Set strong passwords: Teach your child how to create strong passwords that are difficult to guess. Encourage them to use a combination of letters, numbers, and symbols.
2.  Parental awareness: Learn about the latest cyber risks that can affect children, such as cyberbullying, online predators, and inappropriate content.
3.  Use parental controls: Enable parental controls on your child's devices and limit access to certain websites and applications.
4.  Educate your child about online risks: Teach your child about the risks associated with online activities, such as cyberbullying and identity theft.
5.  Monitor online activities: Monitor your child's online activities, suspicious activities, downloads and have regular conversations with them about their online experiences.
6.  Use antivirus software: Install antivirus software on your child's devices to help protect against malware and other cyber threats.
7.  Avoid sharing personal information: Teach your child to never share personal information, such as their name, address, or phone number online.
8.  Report suspicious activities: Teach your child to report any suspicious activities to a trusted adult immediately.

Figure 12 illustrates that by following these safety guidelines, parents and caregivers can help ensure that children are using technology safely and responsibly while minimizing the risks associated with excessive screen time and cybersecurity challenges and safeguarding the child's health.

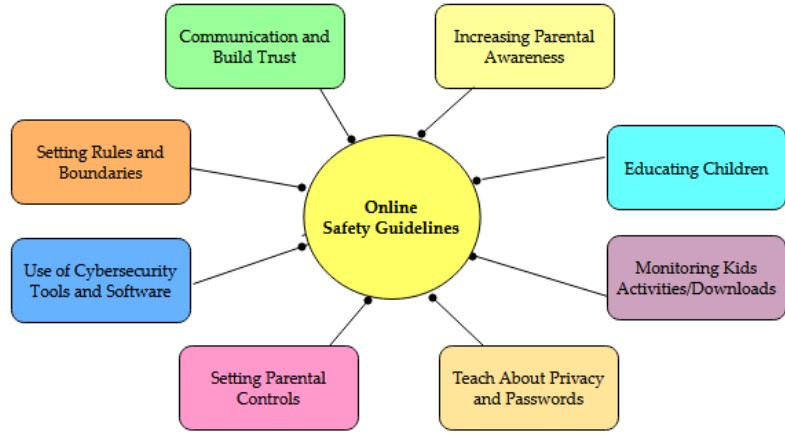

**Figure 12.** Safety Guidelines to Mitigate Cyber Risks in Children.

**Diet and Sleep Affecting Screen Time in Children**

Growing evidence suggests a strong association among screen time, diet, and sleep in children (0–7 years):

**Diet:**

Studies indicate that children who spend more time in front of screens tend to consume more high-calorie, low-nutrient foods, and fewer fruits and vegetables [85]. Children who watch more TV or play more video games are more likely to eat while distracted, which can lead to mindless eating and cause weight-gain problems [86].

**Sleep:**

Research shows that screen time can disrupt children's sleep patterns [40]. Excessive screen usage can cause sleep problems in children and other adverse effects on children's health and development [43]. Too much screen time and regular exposure to poor-quality programming have been linked to inadequate sleep schedules and insufficient sleep in children [87].

It is crucial for parents to be aware of the potential impact of screen time on children's diet and sleep and to take necessary measures to promote healthy habits. Figure 13 offers the suggestions for parents to promote healthy food habits and quality sleep in children [88,89].

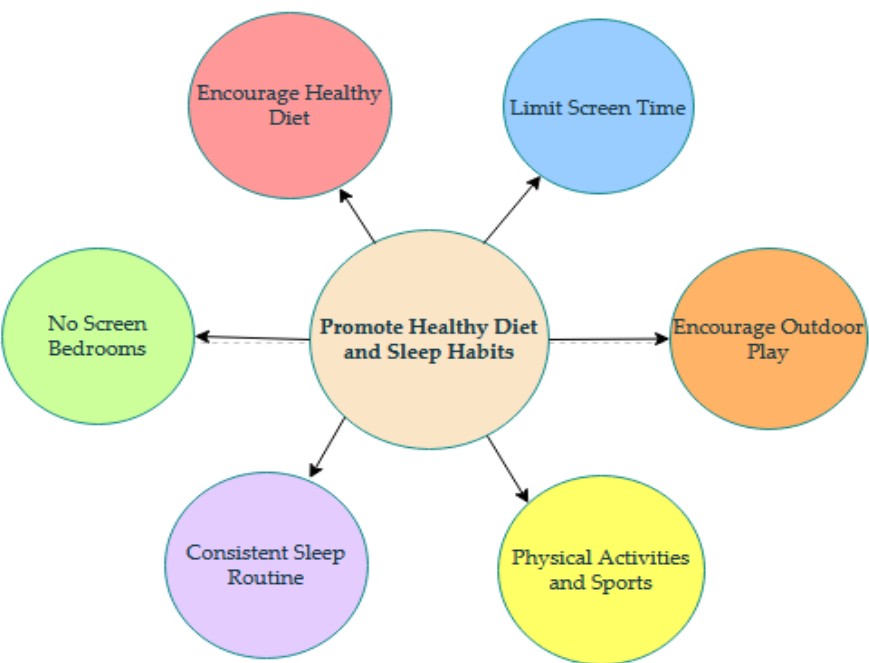

**Figure 13.** Suggestions to parents to promote healthy diet and sleep habits in children.

**Foods That Promote Better Health and Sleep for Children**

Research suggests links between diet and better health/sleep in children. A balanced and nutritious diet can promote overall health and well-being, which may reduce the desire for excessive screen time [89]. Here are some food suggestions that can help promote healthy habits and a balanced lifestyle:

1. Fruits and vegetables: Encourage children to eat a variety of fruits and vegetables, which are rich in vitamins, minerals, and antioxidants. These nutrients can help support brain function and reduce the risk of obesity and other health problems, which may help reduce screen time.

2. Lean protein: Protein is an essential nutrient for growth and development and can help keep children feeling full and satisfied. Lean protein sources such as chicken, fish, beans, and tofu can be included in meals and snacks.

3. Healthy fats: Healthy fats such as those found in nuts, seeds, avocados, and oily fish can help support brain function and reduce inflammation, which may help reduce the desire for excessive screen time.

4.   Complex carbohydrates: Foods such as whole grain bread, brown rice, and oatmeal are rich in fiber and complex carbohydrates, which can help regulate blood sugar levels, reduce cravings for unhealthy snacks, and promote calm and relaxation. This can help promote better sleep in children.

5.   Magnesium-rich foods: Magnesium is a mineral that can help promote relaxation and improve sleep quality. Foods such as leafy green vegetables, nuts, seeds, and whole grains are rich in magnesium and can be included in the diet to promote better sleep.

6.   Calcium-rich foods: Calcium is essential for strong bones and teeth and can also help promote better sleep. Dairy products such as milk, cheese, and yogurt are rich in calcium and can be included in the diet to promote better health and sleep.

7.   Foods rich in tryptophan: Tryptophan is an amino acid that can help promote relaxation and improve sleep quality. Foods such as turkey, chicken, eggs, cheese, and nuts are rich in tryptophan and can be included in the diet to promote better sleep.

8.   Water: Encourage children to drink plenty of water throughout the day, which can help them stay hydrated and alert. Dehydration can lead to fatigue and decreased concentration, which may increase the desire for screen time.

Figure 14 gives a list of foods that can help promote better health and sleep in children. In addition to promoting healthy eating habits, it is important to encourage physical activity, social interaction, and other healthy lifestyle habits to reduce screen time and promote overall health and well-being in children.

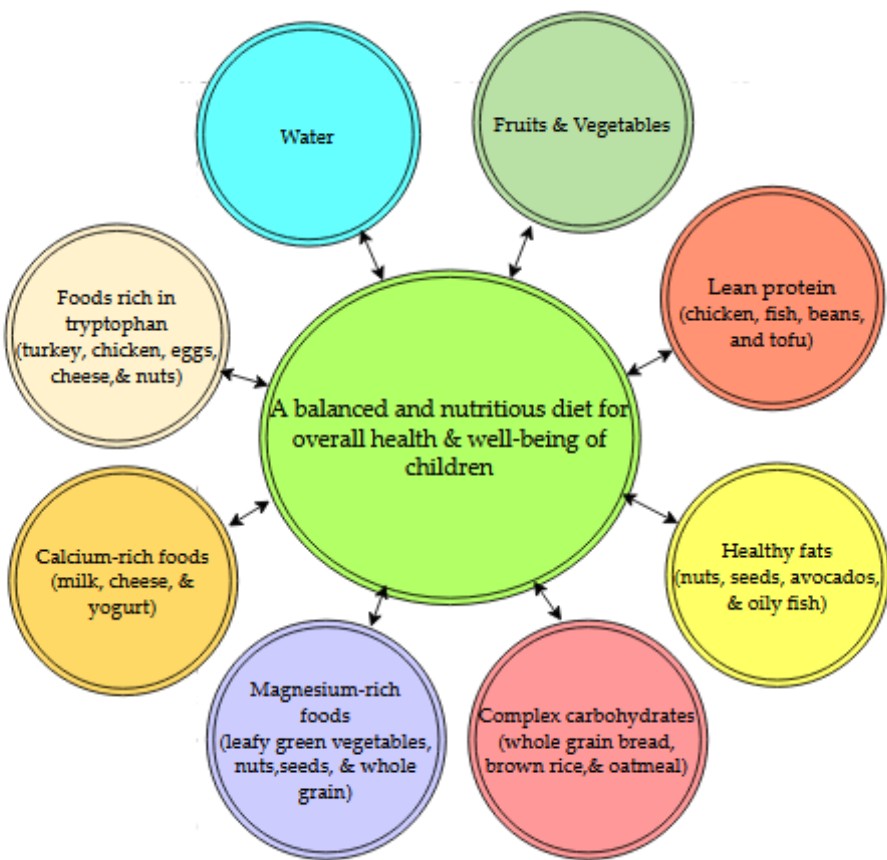

**Figure 14.** Food suggestions to help promote a healthy diet and overall health in children.

**Recommendations for Reducing Screen Time of Children**

Table 9 provides some recommendations for teachers, parents, the community, and kids to reduce children's screen time.

**Table 9.** Recommendations for Reducing Screen Time in Children.

| Teachers | Parents | Community | Kids |
|---|---|---|---|
| 1. Encourage outdoor physical activity. | 1. Enforce screen time limits. | 1. Plan active community events. | 1. Take frequent activity breaks. |
| 2. Plan interactive, screen-free activities. | 2. Offer alternative activities. | 2. Provide non-screen programs. | 2. Limit daily screen time. |
| 3. Limit screen use in class. | 3. Encourage physical activity. | 3. Educate kids to reduce screen time. | 3. Choose non-screen activities. |
| 4. Provide non-screen resources. | 4. Model healthy screen habits. | 4. Partner with schools. | 4. Avoid screens before bed. |
| 5. Assign screen-free homework. | 5. Use parental controls. | 5. Organize parental awareness programs. | 5. Set and track goals. |

## 9. Conclusions

This paper investigated the impact of digital media screen usage on various age groups in children. The investigation reveals that screen time is steadily growing across all age groups: infants, toddlers, preschoolers, and school-age children. Specifically, this paper documents a summary of both potential benefits and detrimental effects of screen time usage across multiple developmental domains: cognitive, physical, language, social, and emotional domains. The positive influences of screen time include enhancing children's learning abilities through educational applications, facilitating video chats with families, accessing educational resources, introducing diverse perspectives, and thus enhancing their creativity and self-expression. The negative influences include technology addiction, stress level increase, reduced physical activities, lack of sleep, emotional distress, relationship issues, and behavioral problems. Although technology and digital media have become ubiquitous in modern society, much remains to be learned about their effects on child development. Further research is needed to explore the complex relationship between screen time and various developmental domains from multiple stakeholders, which can inform evidence-based guidelines and recommendations for healthy media use in children.

Further, the COVID-19 pandemic increased social distancing and remote learning activities and thus increasing the possible potential impacts on child's various developmental domains due to increase in screen time usage. Therefore, it is now more important for parents, teachers, and caregivers to be aware of the potential effects of excessive screen time and take proactive steps to mitigate these risks. However, balancing screen time with other forms of play and interaction is crucial to ensure healthy development in young children. Additionally, various recommendations and guidelines from the AAP and WHO on promoting healthy media use in children are presented. These guidelines stress the importance of limiting screen time, choosing age-appropriate content, and encouraging parental involvement and supervision in children's media use.

## 10. Future Work

Future research could explore parents' best practices and attitudes contributing to healthy screen time habits and their impact on child development domains. Additionally, conducting comprehensive meta-analyses of existing studies could provide a more precise assessment of how the duration of the screen time impacts child development and enable the identification of factors that moderate this relationship. For example, a meta-analysis could investigate whether the type of screen activity, the duration and frequency of screen time, and the child's age play a role in the impact of screen time on development. This information could be analyzed to inform guidelines and recommendations for parents and caregivers on managing their children's screen time to promote healthy development.

**Author Contributions:** The idea for this research was conceptualized from a discussion between the authors V.N.P.-M. and P.R.; while P.R. guided the methodology and formal analysis, the investigation and resource curation were done by V.N.P.-M.; the original draft preparation was done by V.N.P.-M., while the review and editing were performed by V.N.P.-M. under the supervision of P.R. All authors have read and agreed to the published version of the manuscript.

**Funding:** This research work is supported by the Center for Cyber Security Research (C2SR) at the University of North Dakota.

**Institutional Review Board Statement:** Not applicable.

**Informed Consent Statement:** Not applicable.

**Data Availability Statement:** Not applicable.

**Acknowledgments:** The authors acknowledge the support of Patrick J. McCloskey, Dakota Digital Review, North Dakota University System for proofreading this work.

**Conflicts of Interest:** The authors declare no conflict of interest.

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
