# Peer review of "Impact of Screen Time on Children’s Development: Cognitive, Language, Physical, and Social and Emotional Domains"

_mti, doi:10.3390/mti7050052_

Round 1

Reviewer 1 Report

I appreciate the opportunity to review Screentime Impact of Technology on Children (0 -7 Years): A Literature Review; Regarding this research, I have the following suggestions:

1.- The title registered in the platform of the Journal and the title of the pdf is not the same, which is the correct one?

2.- In the title it is convenient to indicate what type of review it is: narrative, comprehensive, systematic.

3.- The abstract should be organized into sections: introduction, methods, results, conclusions... It is NECESSARY TO CONSIDER organizing the entire article following this usual scheme in review papers.

4.- The introduction provides data and references that must be contextualized and indicate whether they are from certain countries or international, for example line 32.

5.- In the introduction it is not usual to put tables, unless they are essential. I think the data from the two tables could be embedded in the text and removed.

6.- The introduction should end with two paragraphs:

- One dedicated to justifying the need for this research, in relation to existing deficiencies in the scientific literature on the subject.

- Another dedicated to indicating the general and specific objectives of this review, objectives that must later be mentioned in the conclusions to indicate whether they have been achieved.

7.- It is necessary to include a section on methods, where it is indicated why this methodology has been chosen, this type of review, how the search for works has been done (inclusion/exclusion criteria, search engines, sources...), how many papers are finally reviewed...etc. A PRISMA type diagram would be appreciated.

8.- In section 2, a citation system is used that is not consistent with the one used in the introduction and that does not facilitate consultation, it is recommended to indicate, at least, the citation number next to the names of the authors, which is sometimes placed at the end of a very long paragraph and it is not known if it corresponds to said authors.

9.- Tables 4 and 5 are not mentioned in the text, therefore it is not known what they are, and what they do there. A single table that summarizes all the articles reviewed is necessary, just as it is done in a review paper.

Author Response

Dear Reviewer,

Thank you for your  review of my manuscript. I appreciate your feedback and comments on the paper.

After carefully reviewing your suggestions, I have made some changes to the manuscript in order to address the concerns you raised. I look forward to hearing back from you and thank you again for your time and expertise.

Please find the attached document for the response and changes.

Best regards,
Vaishnavi

Reviewer 2 Report

Main message of the article

The proposed review emphasizes the impact of screen time exposure on children. The authors discuss findings from the scientific literature according to the domain principally affected (cognitive, linguistic, socio-emotional, physical). The paper also presents the primary recommendation for parental education regarding screen time exposure at the different stages of development of the infant/child.

General Judgment Comments

The topic is fascinating and clinically relevant since screen time, and media exposure are critical issues in public health. The paper is written in a fluent and clear style. The title and abstract report the main subject but can be more informative and reflect better the topic discussed. Furthermore, some information needs to be included in the text; for instance, no “method” section reports how the articles included and discussed were searched and sorted. The results’ presentation respects the scientific standards required but can be improved further. The tables are very informative, same for the figures. In conclusion, some revisions are needed before considering the paper for publication. Please see the comments below.

Major Issues

-       The authors stated that the work presented is a review; however, they must explain what kind of review they are presenting. Therefore, the authors must declare the type of review (systematic, narrative, scoping, umbrella, rapid).

-       Following the previous point, no methodology indicating how the discussed articles were retrieved is described. Please, add a section reporting the specific procedure and criteria adopted to search, find, evaluate, and include the papers discussed in the article’s sections. For further indications, please refer to the Preferred Reporting Items for Systematic reviews and Meta-Analyses (PRISMA) statement (https://www.bmj.com/content/372/bmj.n71)

-       Similarly, how were retrieved and sorted the studies included in Tables 4 and 5? And in which order are they presented? Please, clarify the procedure in the text.

Minor Issues

-       ABSTRACT: in line 15, the authors write “8eightyears of age”; please correct the typo.

-       Line 25: Please report some examples of vital roles played by technology and media in the lives of infants (especially) and toddlers.

-       Are the figures included protected by copyright? If so, please mention in a separate section at the end of the paper that the authors obtained the copyright to use the pictures.

Final comments

The author should edit the article following the comments above and resubmit the manuscript for further consideration.

Author Response

(The authors gave the same response as above.)

Reviewer 3 Report

Review: 

Screentime Impact on Children A Literature Review

Content: 

This paper offers an overview of the potential impacts of technology on young children and discusses how other parents and providers can best manage technology use. 

Introduction: 

"Media before the 2000s was confined just to televisions. With the advent of modern technology, including smartphones, tablets, digital toys, and gaming tools, children are always occupied with digital content that comes in many sizes and forms." The authors compare of technology development. 

The problematic part is: 

"3. Impact of Covid-19 pandemic on children’s screen time"  

I think it is very short and needs to be expanded with some ideas. line 447 - 465, /only 18 lines/ 

for example implement these papers.  These articles make the importance of technology very clear. (and can be inspiring for understanding the harmfulness of technology)  

Tkácová, H.; Pavlíková, M.; Stranovská, E.; Králik, R. Individual (Non) Resilience of University Students to Digital Media Manipulation after COVID-19 (Case Study of Slovak Initiatives). Int. J. Environ. Res. Public Health 202320, 1605. https://doi.org/10.3390/ijerph20021605

Tkacová, H.; Králik, R.; Tvrdoň, M.; Jenisová, Z.; Martin, J.G. Credibility and Involvement of Social Media in Education—Recommendations for Mitigating the Negative Effects of the Pandemic among High School Students. Int. J. Environ. Res. Public Health 202219, 2767. https://doi.org/10.3390/ijerph19052767

I have to appreciate the very well written part:  

How can parents minimize screen time? lines 466 - 502.

I ask the authors to develop the ideas of AAP. This part is only summarized, without author's comments. lines: 504-514 

For the quality of the article, it would be good to develop your own comments. 

 I think that this article has a clear structure, meaning and enriches with new knowledge. 

It would be a shame if the authors did not add their thoughts. 

I recommand to publish this paper  after minor revision 

Author Response

(The authors gave the same response as above.)

Round 2

Reviewer 1 Report

I appreciate that my suggestions have been followed

Reviewer 2 Report

The authors edited the text carefully following the suggestions provided in the first revision. Therefore, the current version of the article can be endorsed for publication.

Good luck with your future work!